# Assessing the Nutritional Impact of an Increase in Orphan Crops in the Kenyan Diet: The Case of Millet

**Cesar Revoredo-Giha** [1,*] **, Hasibi Zavala-Nacul** [2] **and Luiza Toma** [1]

1 Department of Rural Economy, Environment and Society, Scotland's Rural College (SRUC), Edinburgh EH9 3JG, UK; luiza.toma@sruc.ac.uk
2 Independent Food Security Consultant, Mexico City 06000, Mexico; hazana.qa@gmail.com
* Correspondence: cesar.revoredo@sruc.ac.uk

**Abstract:** Orphan crops are those crops that did not receive the same attention of the research community as in the case of staples such as wheat, maize, or rice despite their regional and nutritional importance. A relatively recent trend has been promoting their research to improve their productivity and resilience to environmental shocks. However, their impact on consumers' nutrition has been analysed only considering the crops individually and not in the context of the diet. This is important because an increase in the consumption of one product may trigger changes in the other products that conform to the diet. The purpose of this paper is to assess the potential impact, in terms of food choices and nutrition, of increasing the consumption of orphan crops (represented by millet) in the Kenyan diet. This is carried out using a microeconomic-based methodology, which augments the original consumer problem with a constraint regarding the amount of the orphan crop on the diet. To compute the required elasticities for the method, three demand systems—i.e., for rural, less affluent urban, more affluent urban households—were estimated using the 2015–16 Kenyan Integrated Household Survey and the two-step approach to address the zero consumption for some food categories; the second step was modelled using the Linquad demand model. The results indicate that although the orphan crops have the capacity to improve some of the nutrients (e.g., vitamins and minerals), in net terms, as measured by the aggregated nutritional indicator the improvement is somewhat limited, the improvements occur in the rural and the less affluent population.

**Keywords:** orphan crops; nutrition; healthy consumption





## 1. Introduction

Orphan crops are those crops that have not received the same attention from the research community as in the case of staples such as wheat, maize, or rice despite their regional and nutritional importance. A list of those crops for the case of Africa can be found on the African Orphan Crops Consortium (http://africanorphancrops.org/meet-the-crops/ (accessed on 1 November 2021)) website. A relatively recent trend has been to promote research to improve their productivity and resilience to environmental shocks [1].

Their impact on consumers' nutrition has, however, been analysed only considering the crops individual characteristics [2,3] and not in the context of the diet, where the increase in the consumption of one product may trigger changes in other products that conform to the diet (e.g., [4,5]). This is important because the resulting impact of the increase in the orphan crop is the net nutritional effect on the diet (e.g., [6]).

Kenya, the country of study, has a population of about 51 million inhabitants (by 2018), of which the rural population represents about 73% of the total population. The country is interesting for this case study because its two major consumed cereals according to FAOSTAT are maize (77.5 kg/capita/year) and wheat (38.4 kg/capita/year), none of which the country is self-sufficient in. [7], analysing East and Southern Africa, pointed out that there was a good potential market for sorghum and millet, despite which the millet supply only represents 1.5% of the total supply of maize and wheat together.

The purpose of this paper is to assess the potential impact, in terms of food choices and nutrition, of increasing the consumption of orphan crops (focusing on millet and other minor cereals) on the Kenyan diet by considering three socioeconomic groups (rural, urban less affluent and urban more affluent). This assessment was conducted using a microeconomic-based methodology, which augmented the original consumer problem with a constraint regarding the amount of the orphan crop on the diet.

This study shows the importance to consider consumers' preferences when introducing new products or expanding current products on the diet as the results indicate that the net effect on nutrition is not as impressive as when the introduction of a product is considered in isolation (i.e., not in the context of the diet but individually). In other terms, for instance, if the income is constant, an increase in the orphan crops on the diet will displace other food products; therefore, it is important to take into account the next effect on the diet.

The structure of this paper is as follows: it begins with a literature review; next, the methodology is presented, comprising the simulation method, the estimation of the needed elasticities and the data used for the estimation. Then, the results are discussed. Finally, conclusions are presented.

## 2. Literature Review

The supply of healthy products that respond to consumer preferences can be seen as an effective tool to support healthy diets in situations where consumers face complex choices. This is needed as Africa's consumer markets are showing an expansion of ultra-processed products [8], a fact that could be associated with increasing levels of non-communicable diseases.

The aforementioned ultra-processed foods are displacing more traditional dietary patterns, which are based on fresh and perishable whole or minimally processed foods, some of which are orphan crops. The consumption of these products is more suitable socially, environmentally and nutritionally.

Research has indicated that orphan crops, which are part of traditional diets such as a range of fruits, vegetables, legumes, grains and roots, offer the possibility to support greater food diversification in Africa [1,9,10].

Food diversification [11], which includes the increase in orphan crops as an alternative and is an alternative to the crop biofortification approach, is founded on increasing the range of nutritious crops grown [12] that are available for the farmers. This not only increases food system resilience under variable weather due to less risk and the properties of those crops [13] but also has the possibility to improve nutrition [4,14].

Ref. [3] points out that nutritional wellbeing is important for health and development. Moreover, the nutritional status of a community has therefore been recognized as an important indicator of national development as malnutrition is an impediment in national development and hence assumes the status of a national problem. This requires dietary quality to be taken into consideration. Diversification of food production must be encouraged both at the national and household level in tandem with increasing yields. In the case of millet, it is a nonacid-forming food and is easy to digest. It is considered to be one of the least allergic and most digestible grains available and is a warming grain; it helps to heat the body in cold or rainy seasons. However, the use of finger millet is limited due to the coarse nature of the grain. It has high fibre content and the outer cover of the grain is thick, which makes its processing difficult and gives a poor sensory quality (e.g., [2,3]). However, the full impact of the introduction of millet depends on its relationship with other food products in the diet.

In contrast with approaches that focus solely on the production of orphan crops to increase consumption diversification (e.g., [5]), it is important to note that the impact of orphan crops on consumption and nutrition depends on their uptake by consumers (and potentially to other stakeholders in the supply chain such as processors) because only then these crops can help ensure producers receive a fair and sustainable return for their products, connecting them with markets, itself an effective tool against poverty [15]. For

this, considering consumers' preferences are important. In order to appeal to consumers, orphan crops cannot be a mere afterthought, nor can they simply be framed in terms of development policy or agricultural advantages [16]. As pointed out in the case of millets in India, the potential nutritional, environmental, and economic benefits of embracing agricultural biodiversity are not likely to be enough to change consumers' preferences. It is, therefore, important to improve consumers appreciation for the crops.

Based on the above, the contribution of this paper is to consider the demand for millet and analyse the impact on the Kenyan diet of expanding the quantity of orphan crop cereals on the diet under current consumer preferences (i.e., without the influence of any campaign to increase consumers appreciation for these crops). This provides an assessment of the impact that an increasing amount of orphan crops on the diet may have on nutrition.

## 3. Methodology

This section begins by presenting the evaluation method, followed by the approach to compute the required elasticities; finally, the data are used for the estimation.

### 3.1. Evaluation Method

The approach used in this paper for the ex-ante evaluation of increasing orphan crops on the diet is based on [17]. A brief overview of the method is presented here for the sake of completeness. It is founded on neoclassical consumer theory and assumes that consumers choose the consumption of a bundle of H goods in quantities $q = (q_1, \ldots, q_H)$ to maximise a strictly increasing utility, quasi-concave, twice differentiable utility function $U(q_1, \ldots, q_H)$, subject to a linear budget constraint $p.q \leq M$, where $p$ and $M$ are price and income vectors, respectively.

In this study, the above problem is modified by adding an additional constraint, which is the required level of orphan crops that enter into the diet. Mathematically, the additional constraints (called nutritional constraints in [17]) are expressed by $\sum_{i=1}^{H} a_i^n q_i \leq r_n$, $\forall n = 1, \ldots, N$.

To solve the modified version of the utility maximization problem, the procedure relies on the notion of shadow prices. Duality theory is used to relate the unconstrained Hicksian demand function $h_i(p, U)$ to the constrained model $\widetilde{h}_i(p, U, A, r)$; where $A$ is the N X H matrix of nutritional coefficients and $r$ is the N vector of maximum nutritional amounts.

Shadow prices are calculated by maximizing $\widetilde{C}_i(p, U, A, r)$ subject to $\sum_{i=1}^{H} a_i^n q_i \leq r_n$, $\forall n = 1, \ldots, N$. The Lagrangian of the virtual price problem (L) is expressed by:

$$L = C(\widetilde{p}, U) + \sum_{j=1}^{H} \left(p_j - \widetilde{p}_j\right) h_j + \sum_{n=1}^{N} \mu_n \left(r_n - \sum_{j=1}^{H} a_j^n h_j\right) \tag{1}$$

where $\mu_n$ is the Lagrangian multiplier associated with the nth nutritional constraints. The Kuhn–Tucker conditions for (1) are based on the assumption of non-satiation and strictly positive virtual prices as:

$$\frac{\partial C}{\partial \widetilde{p}_i} - h_i + \sum_{j=1}^{H} \left(p_j - \widetilde{p}_j\right) \frac{\partial h_j}{\partial \widetilde{p}_i} - \sum_{n=1}^{N} \mu_n \sum_{j=1}^{H} a_j^n \frac{\partial h_j}{\partial \widetilde{p}_i} = 0, \ i = 1, \ldots, H \tag{2}$$

$$\mu_n \left(r_n - \sum_{j=1}^{H} a_j^n h_j\right) = 0 \tag{3}$$

$$\mu_n \geq 0, \ n = 1, \ldots, N \tag{4}$$

By applying Shepherd's lemma and replacing $\frac{\partial h_j}{\partial \widetilde{p}_i}$ by $s_{ij}$, Equation (2) reduces to:

$$\sum_{j=1}^{H} \left[ \left( p_j - \widetilde{p}_j \right) - \sum_{n=1}^{N} \mu_n a_j^n \right] s_{ij} = 0, \ i = 1, \ldots, H \tag{5}$$

Assuming that all N equations are binding, the virtual price problem reduces to:

$$\widetilde{p}_i = p_i - \sum_{n=1}^{N} \mu_n a_j^n, \ i, \ldots, \ H \tag{6}$$

$$\sum_{j=1}^{H} a_i^n h_i(\widetilde{p}_i, U) = r_1 \tag{7}$$

According to [17], the first set of Equations (6) implies that deviations between shadow prices and market prices are proportional to the nutritional coefficients of the goods entering the single nutritional constraint. The second set of Equation (7) indicates that the nutritional constraints are binding. A change in the shadow price because of a change in the nutritional constraints can be expressed as:

$$\frac{\partial \widetilde{p}_i}{\partial r_1} = \frac{a_i^1}{\sum_{i=1}^{H} \sum_{i=1}^{H} s_{ij} a_i^1 a_j^1}, \ i, \ldots, H \tag{8}$$

Moreover, a change in the Hicksian demand of product k due to a change in the nutritional constraints is expressed by:

$$\frac{\partial \widetilde{h}_k}{\partial r_1} = \frac{\sum_{i=1}^{H} s_{ki} a_i^1}{\sum_{i=1}^{H} \sum_{i=1}^{H} s_{ij} a_i^1 a_j^1}, \ k = 1, \ldots, H \tag{9}$$

Equations (8) and (9) suggest that a change in the nutritional constraints has an impact on the entire diet of the consumer through substitution and complementary relationships across food products. Equation (9) is used to evaluate how consumers react to a change in the nutritional requirement (in this case, the amount of orphan crop in the diet). As (9) assumes that the level of utility is the same, it is possible that it may exceed the original budget; therefore, it is necessary to compute the change in the Marshallian demands which is given by (10):

$$\Delta x = \Delta h + \widetilde{h} \cdot \varepsilon^R \frac{CV}{p \cdot \widetilde{h}} \tag{10}$$

where $\Delta h = \left( \frac{\partial \widetilde{h}_1}{\partial r_1} \Delta r_1, \ldots \ldots, \frac{\partial \widetilde{h}_K}{\partial r_1} \Delta r_1 \right)$, $\varepsilon^R$ is the vector of income elasticities, CV is the compensating variation which is given by $CV = -p \cdot \Delta h$. Figure 1 presents the flowchart with the simulation procedure.

This study also estimated the change in the nutritional value of the diet due to the inclusion of the orphan crops. This was carried out by computing the mean adequacy ratio (MAR), which estimates the percentage of mean daily intake of beneficial nutrients with 100% representing a diet that would conform to all these nutritional requirements [18]. Note that the components of the MAR are truncated to 100; therefore, excesses of one of the nutrients cannot be compensated for the lack of another nutrient. The formula of the MAR is given by (11), where $c_i$ is the intake of nutrient i, $R_i$ is the recommended intake of nutrient i and m is the number of nutrients.

$$MAR = \frac{1}{m} \times \sum_{i=1}^{m} \frac{c_i}{R_i} \times 100 \tag{11}$$

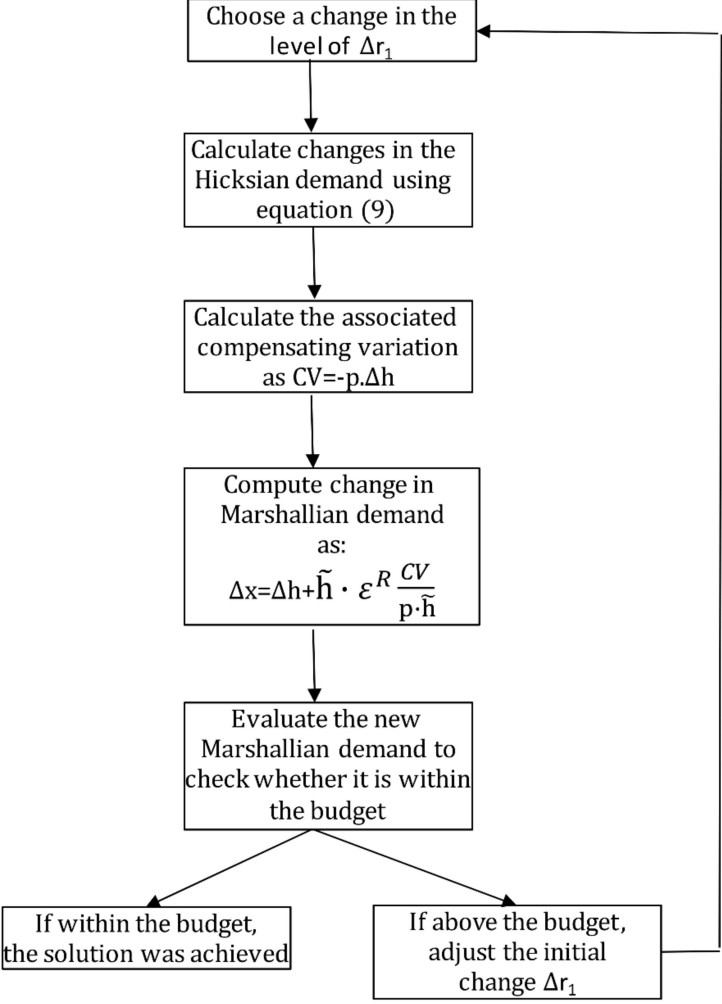

**Figure 1.** Flowchart with the simulation procedure. Source: Own elaboration based on Irz et al. (2015).

### 3.2. Demand Estimation

The assessment of increasing the quantity of orphan crops to the diet while keeping the amount of income constant requires the estimation of demand elasticities. A problem when estimating elasticities using household surveys is the censoring in responses. As pointed out by [19], some households might not consume certain food groups, resulting in a zero value for the dependent variable. This may be due to infrequency of purchase, consumers preferences (i.e., they actually do not consume the food group) or because consumers do not purchase the good at the current prices and income levels (i.e., corner solution).

To address the censoring problem, the estimation was carried out using the two-step procedure [20]. The first step models the zero consumption with a selection mechanism given by (12):

$$\begin{cases} x_i^* = g(p, y) + \epsilon_i \\ d_i^* = z\prime_i \lambda_i + v_i \\ d_i = \begin{cases} 1 \text{ if } d_i^* > 0 \\ 0 \text{ if } d_i^* \leq 0 \end{cases} \\ x_i = d_i \cdot x_i^* \end{cases} \tag{12}$$

where $d_i$ and $x_i$ are the observed values of whether the product i is purchased by the household and the quantity demanded of the product. The '*' indicate latent variables; $g(p, y)$ is a function that depends on prices p and income y; $\epsilon_i$ and $v_i$ are error terms, the $z_i$ are variables affecting the decision of purchase and $\lambda_i$ are the parameters of that function.

The unconditional expected value of system (12) is given by (13):

$$x_i = \phi \cdot g(p, y) + k_i \varphi + \varepsilon_i \tag{13}$$

where $\phi$ and $\varphi$ are the cumulative density and the standard normal density functions, $k_i$ is a parameter and $\varepsilon_i$ is an error term. The $g(p, y)$ needs to be approximated. Here, this paper follows [21] and uses the Linquad demand system [22–24]. The final Marshallian demand specification of the LinQuad model [22] is given by (14):

$$x = \alpha + Av + Bp + \gamma[y - p\prime\alpha - p\prime Av - 0.5p\prime Bp] \tag{14}$$

where $\alpha$, A, B, $\gamma$, $\delta(v)$ are vectors or matrices of parameters and v is a vector of demographic variables. The quadratic term in prices increases the flexibility in Slutsky symmetry removing the restrictions that constrain the preference ordering of a linear system. In addition, the LinQuad quasi-expenditure function is a second-order Taylor series approximation to any arbitrary expenditure function. The prices and income elasticities adjusted by the selection mechanism are given by [21]:

$$\eta_{ij}^a = \phi_i \eta_{ij}^s + \frac{\varphi_i \lambda_{ij} p_j}{x_i} \{x_i - \kappa_i(z\lambda)\} \tag{15}$$

$$\eta_i^a = \phi_i \eta_i^s + \frac{\varphi_i \lambda_{iy} y}{x_i} \{x_i - \kappa_i(z\lambda)\} \tag{16}$$

where the original elasticities (not considering the selection mechanism) are given by:

$$\eta_{ij}^s = \left( \beta_{ij} - \gamma_i (\alpha_j + A_j v + B_j p) \right) \frac{p_j}{x_i} \tag{17}$$

$$\eta_i^s = \gamma_i \frac{y}{x_i} \tag{18}$$

The Hicksian elasticities were computed using the Slutzky formula $\eta_{ij}^{a*} = \eta_{ij}^a + \omega_j \eta_i^a$. The unconditional elasticities for the cereals and pulses category were computed using the formulas by [25].

### 3.3. Data Used in the Analysis

The data used came from the 2015/16 Kenya Integrated Household Budget Survey (KIHBS) which was conducted over a 12-month period to obtain up-to-date information on a range of socioeconomic indicators used to monitor the implementation of development initiatives [26].

The survey collected information on household characteristics, housing conditions, education, general health characteristics, nutrition, household income and credit, household transfers, information communication technology, domestic tourism, shocks to household welfare and access to justice.

The KIHBS 2015/16 is a multi-indicator survey with the main objective of updating the household consumption patterns in all counties. KIHBS 2015/16 is designed to provide estimates for various indicators at the county level. A total of 50 study domains are envisaged. These are: all forty-seven (47) counties (each as a separate domain), urban and rural (each as a separate domain at the national level), and lastly the national-level aggregate.

The sample for KIHBS 2015/16 is a stratified sample selected in two stages from the master sample frame. Stratification was achieved by separating each county into urban and rural areas; in total, 92 sampling strata were created since Nairobi County and Mombasa County have only urban areas. Samples were selected independently in each sampling stratum by a two-stage selection. In total, the national sample size for KIHBS 2015/16 comprised a total of 23,880 households from 2388 clusters. Note that the actual dataset has 21,754 observations after cleaning.

The 2015/16 KIHBS data were weighted to be representative at the national level as well as at the county level. The weighting was based on the selection probabilities in each domain. The design weights were adjusted using the survey response to give the final weights.

The survey collected information about consumption and expenditure on food items, regular non-food items and durable goods and services. The data comprised food purchased, net received, in stock in terms of quantities and expenditure. Food information was grouped as shown in Figure 2, showing orphan crops (millet grain, millet flour, cassava flour, sorghum grain, sorghum, flour, sesame seeds and mixed porridge flour). Note that the most important component of the orphan crop group was millet grain.

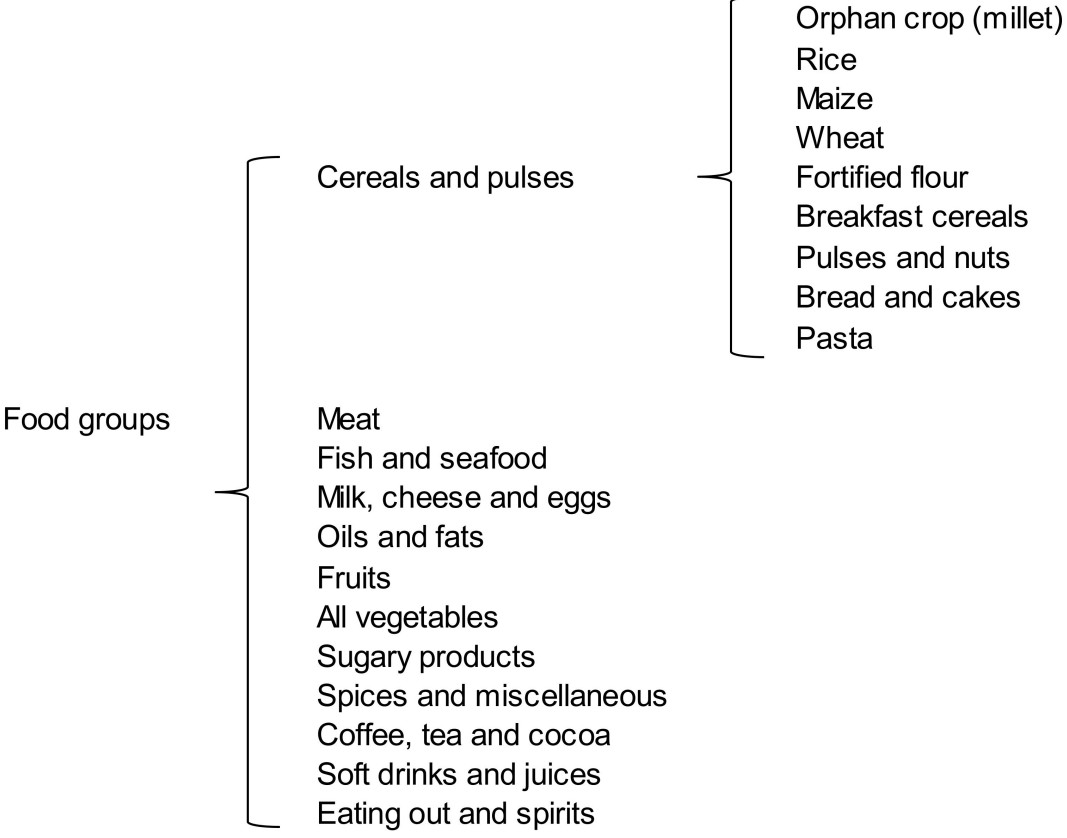

**Figure 2.** Food groups for the analysis.

As shown in Figure 3, the socioeconomic groups considered for the analysis were: rural (13,079 households), urban less affluent (5784 households) and urban more affluent (2895 households). The less and more affluent households were set based on total expenditure quintiles (i.e., the households in the lowest three urban expenditure quintiles were classified as the least affluent). Table 1 provides information about the per capita consumption of the different groups and Table 2 shows data for several social variables by group.

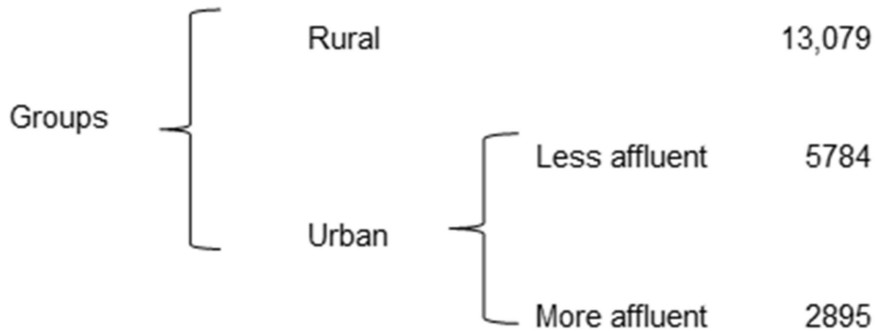

**Figure 3.** Socioeconomic groups for the analysis. Source: Own elaboration based on the 2015/16 KIHBS. Note: The numbers on the figures are the total households in the group.

**Table 1.** Descriptive statistics for each one of the socioeconomic groups in adult equivalent (units are in the note 1/).

| | **Rural** | | | **Urban Poor** | | | **Urban Affluent** | | |
|---|---|---|---|---|---|---|---|---|---|
| | **Quantity** | **Price** | **Zero (%)** | **Quantity** | **Price** | **Zero (%)** | **Quantity** | **Price** | **Zero (%)** |
| Orphan crop | 0.153 | 17.952 | 21.6 | 0.057 | 12.707 | 13.0 | 0.059 | 15.263 | 11.2 |
| Rice | 0.368 | 56.763 | 62.0 | 0.430 | 70.844 | 75.2 | 0.655 | 95.235 | 80.1 |
| Maize | 2.276 | 42.752 | 92.0 | 1.365 | 39.636 | 79.9 | 1.012 | 38.182 | 64.6 |
| Wheat | 0.143 | 15.913 | 22.1 | 0.142 | 15.953 | 23.1 | 0.188 | 18.146 | 27.0 |
| Fortified flour | 0.307 | 24.366 | 33.3 | 0.407 | 32.125 | 45.8 | 0.683 | 39.980 | 58.6 |
| Breakfast cereals | 0.001 | 2.503 | 0.4 | 0.000 | 4.882 | 0.6 | 0.011 | 45.956 | 6.7 |
| Pulses and nuts | 0.581 | 72.183 | 79.8 | 0.318 | 66.425 | 63.7 | 0.391 | 89.838 | 65.0 |
| Bread and cakes | 0.227 | 81.579 | 59.9 | 0.344 | 95.587 | 70.8 | 0.815 | 122.052 | 87.3 |
| Pasta | 0.009 | 4.830 | 2.9 | 0.020 | 13.352 | 8.0 | 0.059 | 40.911 | 21.6 |
| Meat | 0.228 | 176.825 | 50.6 | 0.181 | 217.849 | 42.3 | 0.606 | 320.333 | 19.6 |
| Fish and seafood | 0.083 | 116.944 | 66.8 | 0.069 | 135.475 | 65.3 | 0.122 | 155.006 | 65.1 |
| Milk, cheese and eggs | 1.758 | 69.783 | 11.3 | 1.241 | 90.492 | 11.7 | 2.289 | 99.738 | 6.3 |
| Oils and fats | 0.168 | 175.329 | 5.5 | 0.188 | 161.938 | 7.2 | 0.299 | 179.998 | 8.2 |
| Fruits | 0.900 | 38.732 | 28.5 | 0.742 | 53.706 | 19.1 | 1.792 | 69.315 | 6.1 |
| All vegetables | 2.872 | 43.954 | 6.0 | 2.196 | 54.580 | 5.1 | 3.787 | 54.573 | 6.5 |
| Sugary products | 0.631 | 95.173 | 5.3 | 0.394 | 99.895 | 6.8 | 0.449 | 121.626 | 8.2 |
| Spices and miscellaneous | 0.052 | 63.559 | 3.9 | 0.045 | 70.325 | 7.3 | 0.074 | 110.950 | 8.5 |
| Coffee, tea and cocoa | 0.024 | 489.460 | 7.4 | 0.024 | 522.454 | 10.4 | 0.044 | 549.160 | 11.4 |
| Soft drinks and juices | 0.113 | 20.442 | 80.2 | 0.127 | 23.098 | 77.0 | 0.912 | 51.226 | 44.4 |
| Eating out and spirits | 0.244 | 59.804 | 81.4 | 0.238 | 50.992 | 84.6 | 0.722 | 255.130 | 73.0 |
| Total expenditure | 1232.3 | | | 1415.2 | | | 4069.1 | | |

Note 1/Units in kg or lts per week per adult equivalent; prices are in Kenyan shillings per kg or lts. Total expenditure is in Kenyan shillings per week. Zero indicates the percentage of households with positive consumption in the group. Source: Own elaboration based on the 2015/16 KIHBS.

Table 1 shows that the per capita consumption of maize in rural areas is almost twice the amount in urban areas (with the less affluent urban group being close to the rural group). In all the areas, maize appears more important than other cereals such as rice and wheat. In addition, clearly, the orphan crop (i.e., millet) is more important in rural than in urban areas. The consumption of vegetables is higher in affluent urban areas, although rural and less affluent urban areas also showed a high quantity (about 2/3 of the affluent urban consumption). The other point that is clear from Table 1 is that the consumption of animal protein (dairy products and meat) increases with income.

The nutritional information used in the analysis was from the Kenyan Food Composition Tables (KFCT) [27]. These tables provide very disaggregated data (energy, macronutrients and micronutrient) for a high number of food products.

**Table 2.** Characteristics of each socioeconomic group.

| Category | Rural | Urban | |
| --- | --- | --- | --- |
| | | Less Affluent | More Affluent |
| **Total household members** | | | |
| Less or equal than 3 | 0.37 | 0.47 | 0.74 |
| From 4 to 6 persons | 0.43 | 0.41 | 0.24 |
| From 7 to 10 persons | 0.19 | 0.11 | 0.02 |
| More than 10 people | 0.02 | 0.01 | 0.00 |
| **Gender of head of household** | | | |
| Female | 0.36 | 0.29 | 0.27 |
| Male | 0.64 | 0.71 | 0.73 |
| **Number of children (lesser or equal than 10 years old)** | | | |
| None | 0.34 | 0.41 | 0.63 |
| Less than 3 | 0.56 | 0.54 | 0.36 |
| More than 3 and less than 7 | 0.10 | 0.05 | 0.01 |
| More than 7 | 0.00 | 0.00 | 0.00 |
| **Number of older (greater than 65 years old)** | | | |
| None | 0.34 | 0.41 | 0.63 |
| Less than 3 | 0.56 | 0.54 | 0.36 |
| More than 3 | 0.10 | 0.05 | 0.01 |
| **Type of dwelling** | | | |
| Bungalow | 0.76 | 0.39 | 0.18 |
| Flat | 0.01 | 0.08 | 0.35 |
| Landhi | 0.06 | 0.30 | 0.28 |
| Maisonnette | 0.00 | 0.01 | 0.03 |
| Manyatta/traditional house | 0.14 | 0.03 | 0.00 |
| Shanty | 0.01 | 0.03 | 0.02 |
| Swahili | 0.03 | 0.16 | 0.13 |
| Not stated or other | 0.00 | 0.01 | 0.01 |
| **Activity** | | | |
| Apprentice | 0.00 | 0.00 | 0.00 |
| Contributing family worker | 0.02 | 0.00 | 0.00 |
| Members of producers? cooperatives | 0.00 | 0.00 | 0.00 |
| Own-account worker | 0.55 | 0.33 | 0.31 |
| Paid employee (within hh) | 0.15 | 0.19 | 0.11 |
| Paid employee (outside hh) | 0.19 | 0.38 | 0.49 |
| Working employer . . . | 0.00 | 0.00 | 0.01 |
| Volunteer or Other | 0.09 | 0.09 | 0.08 |
| **Area** | | | |
| Government related | 0.08 | 0.10 | 0.12 |
| Private sector | 0.23 | 0.34 | 0.44 |
| NGO related | 0.01 | 0.01 | 0.02 |
| Agricultural pastoralist | 0.42 | 0.10 | 0.03 |
| Informal sector | 0.05 | 0.14 | 0.08 |
| Other | 0.10 | 0.10 | 0.08 |
| **Richest counties** | | | |
| Nairobi | 0.00 | 0.20 | 0.42 |
| Kiambu | 0.03 | 0.08 | 0.09 |
| Nyeri | 0.03 | 0.02 | 0.02 |
| Kajiado | 0.02 | 0.03 | 0.03 |
| Nakuru | 0.05 | 0.06 | 0.06 |
| Kwale | 0.02 | 0.01 | 0.01 |
| Likipia | 0.02 | 0.01 | 0.01 |
| Murang'a | 0.04 | 0.02 | 0.01 |
| Mombasa | 0.00 | 0.07 | 0.09 |
| Machakos | 0.02 | 0.05 | 0.04 |
| **Poorest counties** | | | |
| Mandera | 0.01 | 0.01 | 0.00 |
| Bomet | 0.02 | 0.01 | 0.00 |
| Elgeyo/Marakwet | 0.01 | 0.00 | 0.00 |
| Samburu | 0.01 | 0.00 | 0.00 |
| West Pokot | 0.02 | 0.00 | 0.00 |
| Migori | 0.03 | 0.01 | 0.00 |
| Turkana | 0.02 | 0.03 | 0.01 |
| Busia | 0.02 | 0.01 | 0.00 |
| Baringo | 0.02 | 0.01 | 0.01 |
| Homa Bay | 0.02 | 0.02 | 0.01 |

## 4. Results and Discussion

Tables 3 and 4 present the baseline and results of the simulations by economic group. The simulations consisted of increasing the amount of orphan crops in the group diet—two,

three and four times. Note that these increases are not as large as they may appear due to the fact that the amount of cereal orphan crops in the diets was very small.

**Table 3.** Percentage changes in the diet composition due to increases in orphan crops by socioeconomic group.

| | Rural | | | | Urban—Less Affluent | | | | Urban—More Affluent | | | |
|---|---|---|---|---|---|---|---|---|---|---|---|---|
| | Baseline 1/ | Simulations 2/ | | | Baseline 1/ | Simulations 2/ | | | Baseline 1/ | Simulations 2/ | | |
| | | 2 Times | 3 Times | 4 Times | | 2 Times | 3 Times | 4 Times | | 2 Times | 3 Times | 4 Times |
| Orphan crop | 0.153 | 100.00 | 199.99 | 300.00 | 0.057 | 100.00 | 200.00 | 299.99 | 0.059 | 100.00 | 200.00 | 300.01 |
| Rice | 0.368 | −5.34 | −10.67 | −3.87 | 0.430 | −3.30 | −4.36 | −4.89 | 0.655 | −1.49 | −2.98 | −4.47 |
| Maize | 2.276 | 0.50 | 1.00 | 0.38 | 1.365 | 0.82 | 1.09 | 1.23 | 1.012 | −0.95 | −1.90 | −2.85 |
| Wheat | 0.143 | −6.01 | −12.03 | −4.35 | 0.142 | −3.13 | −4.14 | −4.65 | 0.188 | −1.92 | −3.84 | −5.75 |
| Fortified flour | 0.307 | −7.03 | −14.05 | −5.06 | 0.407 | −2.28 | −3.03 | −3.40 | 0.683 | 1.86 | 3.72 | 5.58 |
| Breakfast cereals | 0.001 | −57.14 | −100.00 | −55.96 | 0.000 | −68.62 | 62.34 | 196.72 | 0.011 | −13.05 | −26.11 | −39.16 |
| Pulses and nuts | 0.581 | 0.77 | 1.54 | 0.57 | 0.318 | −1.47 | −1.97 | −2.22 | 0.391 | 2.47 | 4.94 | 7.41 |
| Bread and cakes | 0.227 | −4.73 | −9.47 | −3.45 | 0.344 | −2.74 | −3.63 | −4.09 | 0.815 | −1.84 | −3.68 | −5.52 |
| Pasta | 0.009 | −30.99 | −61.97 | −22.88 | 0.020 | −1.75 | −2.33 | −2.63 | 0.059 | 0.39 | 0.78 | 1.17 |
| Meat | 0.228 | −0.02 | −0.05 | −0.18 | 0.181 | 0.13 | 0.09 | 0.01 | 0.606 | 0.01 | 0.01 | 0.02 |
| Fish and seafood | 0.083 | −0.02 | −0.04 | −0.15 | 0.069 | 0.19 | 0.14 | 0.03 | 0.122 | 0.01 | 0.01 | 0.02 |
| Milk, cheese and eggs | 1.758 | −0.02 | −0.03 | −0.11 | 1.241 | 0.13 | 0.09 | 0.01 | 2.289 | 0.00 | 0.00 | 0.01 |
| Oils and fats | 0.168 | −0.01 | −0.02 | −0.09 | 0.188 | 0.09 | 0.06 | 0.01 | 0.299 | 0.00 | 0.01 | 0.01 |
| Fruits | 0.900 | −0.02 | −0.03 | −0.12 | 0.742 | 0.10 | 0.07 | 0.01 | 1.792 | 0.00 | 0.01 | 0.01 |
| All vegetables | 2.872 | −0.02 | −0.03 | −0.12 | 2.196 | 0.12 | 0.09 | 0.01 | 3.787 | 0.00 | 0.00 | 0.01 |
| Sugary products | 0.631 | −0.01 | −0.03 | −0.09 | 0.394 | 0.07 | 0.05 | 0.00 | 0.449 | 0.00 | 0.00 | 0.00 |
| Spices and miscellaneous | 0.052 | −0.01 | −0.02 | −0.06 | 0.045 | 0.06 | 0.04 | 0.00 | 0.074 | 0.00 | 0.01 | 0.01 |
| Coffee, tea and cocoa | 0.024 | −0.02 | −0.03 | −0.11 | 0.024 | 0.08 | 0.06 | 0.01 | 0.044 | 0.00 | 0.01 | 0.01 |
| Soft drinks and juices | 0.113 | −0.03 | −0.06 | −0.23 | 0.127 | 0.24 | 0.18 | 0.04 | 0.912 | 0.01 | 0.02 | 0.02 |
| $P_0$ and $\Delta P$ on the orphan crop 2/ | 30.05 | −34.37 | −68.74 | −25.78 | 24.10 | −12.45 | −16.59 | −18.66 | 29.96 | −25.70 | −51.39 | −77.09 |

Note 1/Units in kg or lts; 2/Changes (%) with respect to the baseline except for prices.

As shown in Table 3, the increase in orphan crops in the diet has effects on all the goods but these changes (note that they are in percentages) differ by food group and socioeconomic group. Moreover, in order to enter those proportions in the diet, the price is required to decrease significantly (as shown by the changes in shadow prices). Recall that these changes are due to the relationship between the different food and drink categories and behind this are preferences, prices and the fact that all the products compete for income.

In Table 3, most of the changes due to the expansion of orphan crops occur within the 'cereals and pulses' category (i.e., its own category). Within the rural households, the increase in orphan crops brings an increase in maize, pulses and nuts. All the other food categories showed a decrease. Note that if the increase in millet had been taken in isolation (without considering potential substitutions), all the negative changes in Table 3 would have been zero (i.e., they would have remained as in the baseline).

Urban households had quite different responses compared to rural households; the most important one of these being the positive response of the non-cereal and pulses response. This is, of course, due to the different sets of elasticities.

In the case of the less affluent urban households, they show, like rural households, an increase in the presence of maize on the diet as more orphan crops are included. Another change is the increase in breakfast cereals (although the quantities are very small) and only when the quantity of millet is duplicated. All the other products within cereals and pulses show a decrease in their quantities.

The most affluent urban households show quite a different response than the other two socioeconomic groups. In contrast with the other two groups, the quantity of maize in the diet is reduced. However, there is an increase in fortified flour, pulses and nuts and pasta. All the other foods within the cereal and pulses categories show a decrease with respect to the baseline.

**Table 4.** Percentage change in nutrients due to increases in orphan crops by socioeconomic group.

| | Rural | | | | Urban—Less Affluent | | | | Urban—More Affluent | | | |
| | Baseline 1/ | Simulations 2/ | | | Baseline 1/ | Simulations 2/ | | | Baseline 1/ | Simulations 2/ | | |
| | | 2 Times | 3 Times | 4 Times | | 2 Times | 3 Times | 4 Times | | 2 Times | 3 Times | 4 Times |
|---|---|---|---|---|---|---|---|---|---|---|---|---|
| Energy (kcal) | 2360.82 | 1.69 | 3.37 | 6.06 | 1828.07 | 0.63 | 1.52 | 2.47 | 2921.79 | 0.30 | 0.59 | 0.89 |
| Protein (g) | 67.93 | 2.46 | 4.92 | 8.17 | 49.95 | 1.01 | 2.29 | 3.62 | 87.69 | 0.69 | 1.38 | 2.07 |
| Lipid total (g) | 62.71 | 1.31 | 2.61 | 4.56 | 55.92 | 0.33 | 0.86 | 1.43 | 98.62 | 0.19 | 0.37 | 0.56 |
| Carbohydrate (g) | 350.03 | 1.57 | 3.14 | 5.99 | 258.96 | 0.62 | 1.56 | 2.56 | 378.81 | 0.22 | 0.44 | 0.66 |
| Fibre (g) | 59.23 | 3.19 | 6.39 | 9.72 | 41.03 | 1.66 | 3.38 | 5.11 | 57.35 | 1.14 | 2.28 | 3.43 |
| Calcium (g) | 1074.43 | 3.22 | 6.45 | 10.05 | 794.37 | 1.61 | 3.24 | 4.88 | 1422.85 | 0.99 | 1.97 | 2.96 |
| Iron (mg) | 28.60 | 4.91 | 9.83 | 15.18 | 21.01 | 2.40 | 4.93 | 7.48 | 35.97 | 1.48 | 2.96 | 4.44 |
| Zinc (mg) | 12.59 | 2.61 | 5.22 | 8.33 | 8.90 | 1.26 | 2.66 | 4.10 | 14.12 | 0.82 | 1.63 | 2.45 |
| Magnesium (mg) | 479.68 | 5.28 | 10.56 | 16.22 | 334.81 | 2.64 | 5.46 | 8.32 | 506.71 | 1.93 | 3.85 | 5.78 |
| Phosphorus (mg) | 1650.31 | 2.84 | 5.68 | 9.06 | 1185.57 | 1.35 | 2.86 | 4.40 | 1922.71 | 0.83 | 1.66 | 2.49 |
| Potassium (mg) | 3812.10 | 2.25 | 4.51 | 6.91 | 2813.32 | 1.07 | 2.18 | 3.29 | 4879.13 | 0.69 | 1.38 | 2.08 |
| Sodium (mg) | 901.82 | −0.53 | −1.06 | −0.17 | 780.94 | −0.50 | −0.58 | −0.59 | 1601.54 | −0.48 | −0.95 | −1.43 |
| Selenium (mcg) | 64.06 | 4.11 | 8.23 | 14.10 | 48.21 | 1.73 | 3.95 | 6.28 | 86.91 | 1.03 | 2.05 | 3.08 |
| Vitamin C (mg) | 169.15 | 1.28 | 2.57 | 3.78 | 129.45 | 0.72 | 1.32 | 1.88 | 238.95 | 0.37 | 0.74 | 1.12 |
| Thiamin—Vitamin B1—(mg) | 1.69 | 2.33 | 4.65 | 7.60 | 1.23 | 1.01 | 2.27 | 3.57 | 1.92 | 0.67 | 1.35 | 2.02 |
| Riboflavin—Vitamin B2—(mg) | 1.85 | 1.11 | 2.22 | 3.93 | 1.38 | 0.51 | 1.14 | 1.78 | 2.55 | 0.28 | 0.56 | 0.85 |
| Niacin—Vitamin B3—(mg) | 13.23 | 2.57 | 5.14 | 8.75 | 10.13 | 1.06 | 2.44 | 3.88 | 17.61 | 0.58 | 1.15 | 1.73 |
| Vitamin A—(µg retinol equivalent) | 2459.53 | 0.60 | 1.20 | 1.81 | 2177.80 | 0.35 | 0.58 | 0.79 | 4720.73 | 0.11 | 0.23 | 0.34 |
| Folate (µg dietary folate equivalence) | 523.75 | 2.80 | 5.61 | 8.62 | 373.44 | 1.29 | 2.75 | 4.23 | 597.42 | 1.01 | 2.02 | 3.04 |
| Vitamin B12 (µg retinol equivalent) | 5.07 | −0.18 | −0.36 | −0.27 | 4.03 | −0.01 | −0.10 | −0.20 | 10.91 | −0.04 | −0.09 | −0.13 |
| $MAR_0$ and $\Delta MAR$ 2/ | 92.03 | 0.16 | 0.32 | 0.55 | 85.41 | 0.28 | 0.58 | 0.90 | 96.96 | −0.03 | −0.07 | −0.10 |

Note: 1/Units in kg or lts; 2/Changes with respect to the baseline except for the baseline MAR.

Table 4 shows two aspects: first, the increases in orphan crops that have an effect on food choice (Table 3) also have an effect on the diet and overall nutrition (which is different from the change in the orphan crop in isolation). Second, in contrast with the observed food choice responses, the nutritional results are qualitatively very similar amongst all the socioeconomic groups. All three groups show a decrease in sodium and a decrease in vitamin B12; otherwise, all the other nutrients show an increase.

It is important to note that these results are not only because of the increased quantity of orphan crops on the diet but also the changes in the other products. These results are due to the interaction of all the products on the diet; in this sense, consumers' preferences, as well as prices and incomes, are key elements influencing consumers' choices.

Whilst the reduction in sodium is welcomed, the small decrease in vitamin B12 can potentially cause severe and irreversible damage, especially to the brain and nervous system [28].

As pointed out in [28], at levels only slightly lower than normal, people, especially those over 60, may feel a range of symptoms such as fatigue, difficulty walking, depression, poor memory, breathlessness, headaches among others. Moreover, the main type of vitamin B12 deficiency anaemia is pernicious anaemia.

The aggregated indicator of nutrition adequacy (MAR) indicates that for the rural and less affluent urban groups, the MAR benefits from the increase in orphan crops on the diet; whilst in the case of the more affluent group, the effect is the opposite. This indicates that the increase in orphan crops on the diet has a particular effect on poorer groups. In addition, since the more affluent urban group is the one most affected by issues related to a westernised diet, it would pay to conduct further research on how orphan crops consumption can replace ultra-processed foods [29].

Orphan crops, as mentioned earlier, are relevant for various reasons. They are linked to the biocultural heritage and to climate resilience. However, to better understand the effect of their promotion and inclusion in the nutritional targets of the population, it is important to revise their impact in the complete diet of the different socioeconomic sectors. One example is the differentiated impact that the increasing consumption of orphan crops has on key food groups such as fruits and vegetables for each sector. In the rural socio-

economic group, fruits and vegetables are reduced but the overall diet adequacy is not necessarily compromised as shown by the improvement in the MAR results. While, for the less affluent urban group, the consumption of these two food groups is positively impacted and so too is the MAR score. With the creation of national prioritization committees in Kenya that focus on the inclusion of orphan crops to achieve the nutritional targets and diversified diets [30], the role of studies that analysed the possible impact of dietary recommendations on the food choices of remaining households for different population sectors helps in understanding the consequences of policy recommendations and can point to better utilization strategies and map risks based on the results. In this scenario, one key risk factor to consider when seeking to double the intake of orphan crops is the decrease in vitamin B12. The results also show that orphan crop inclusion and recommendations in the food systems should be carried out following sector-based dietary guidelines to tackle the differentiated ramifications in the food choices of each group.

## 5. Conclusions

The purpose of this paper has been to assess the potential impact, in terms of food choices and nutrition, of increasing the consumption of orphan crops (using millet as an example of an orphan crop) on the Kenyan diet. This was carried out using a microeconomic-based methodology, which augments the original consumer problem with a constraint regarding the amount of orphan crop on the diet.

This is important because there is increasing interest to promote research to improve orphan crops' productivity and resilience to environmental shocks; whilst their impact on consumers' nutrition has, however, been analysed only considering the crops' individual characteristics and not in the context of the diet.

The overall results indicate that given the current preferences (as measured by the demand elasticities), increasing orphan crops in the diet requires a significant decrease in their price. If this is achieved, the inclusion of more orphan crops can improve the nutritional situation of rural and less affluent households (as measured by the MAR) and worsen the situation of the most affluent households.

The results also indicate that if the role of orphan crops is to be expanded, there will be a need to develop not only the supply in isolation but also (in parallel) the demand for those crops because of productivity gains, which will reduce the cost of production; moreover, prices would need to be compensated by a significant increase in the demand.

It is expected that the above results will be of interest to researchers interested in testing the expansion of healthy products in the context of the diet and to test the net nutritional results. On the one hand, this approach may provide results that are less spectacular than when the product is considered in isolation; however, on the other hand, the results are more realistic as it considers the interaction with other products in the diet. Additionally, these results become relevant for prioritization committees that could use the model for policy recommendations and communication strategies to achieve high-quality diets to comply with nutritional targets, guiding researchers and representatives to map key opportunities and incentives that lead to positive shifts in the food systems.

An interesting extension to the above methodology would be to consider the case of household models, i.e., the case when production and consumption decisions cannot be disentangled. This could also consider the effect of the food markets (e.g., influencing the decisions of consuming or selling the produced products).

**Author Contributions:** Data curation, H.Z.-N.; Formal analysis, C.R.-G. and H.Z.-N.; Investigation, L.T.; Methodology, C.R.-G.; Project administration, C.R.-G.; Visualization, L.T.; Writing—original draft, C.R.-G.; Writing—review & editing, H.Z.-N. and L.T. All authors have read and agreed to the published version of the manuscript.

**Funding:** This research was funded by Biotechnology and Biological Sciences Research Council, grant number [BB/P022537/1].

**Institutional Review Board Statement:** Not applicable.

**Informed Consent Statement:** Not applicable.

**Data Availability Statement:** The data are available from the authors upon request.

**Conflicts of Interest:** The authors declare no conflict of interest.

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
