# Peer review of "Assessing the Nutritional Impact of an Increase in Orphan Crops in the Kenyan Diet: The Case of Millet"

_sustainability, doi:10.3390/su14052704_

Round 1
Reviewer 1 Report
RECCOMENDATIONS In the whole text:
term orphan crops should be reconsidered in all the text because it is not correct in agricultural methodology
Page 1 title to be changed
RECONSIDER THE TITLE the part impact on increase of orphan crops is awkwardly set
also orphan crops might not be methodologically or agriculturaly correct
also crops are a group of cereals and canot be put together in analysis- i t is metodologically wrong
you cannot compare rice as one product and then pasta as one product because pasta is a group of products
what you did is a comparison of population groups and their use of food groups
Abstract should be tottaly changed, the sentences have no flow, meanings are a unclear,
Line 8-14 you are skipping the meaning and jump from environmental schocks to consumers and jump to diet with no logical sense, links or explanations
line 10 a sentence should be put here
since from the global perspective you switch to resilience of orphan crops -name them in the world not only in Kenya exist orphan crops
indicate more reasons or geographical importance (state clear, measurable data)
then forward with nutritional impact (state clear, measurable data)
in the introduction exact evidence, found evidence , findings, should be put in the text, not desriptions (state clear, measurable data)
to be described which crops in which country, if it is Kenya describe if not state which country which crops (state clear, measurable data)
more literature of other authors should (state clear, measurable data) be used ,
start from global perspective,
global crops (state clear, measurable data)
dietery use of crops and then orphan crops (state which)
then Kenyan perspective, (state clear, measurable data)
Kenyan crops, (state clear, measurable data)
Then initiatite consumers perefrences, firts worlwide, than maybe Africa, then Kenya (state clear, measurable data)
Finnaly introduce orphan crops (state which)
Then nutritional value of orpah crops (state clear, measurable data)
Then use in the Diet (state clear, measurable data)
LITERATURE REVIEW
Very confusing and unreadable review
The start should be to introduce the readers about orpahn crops, whic are they , how much is producesd in tons hectares for example at least in Africa, then for Kenya
Then demographic is used in claculation with no evidence what the population is in Kenya,
Further no data about diets not about consumer behaviour in Kenya toward food is in the text
Since the paper refers to mathematical countings and formulas some evidence should be set from the findings of other author, then hich crops were taken in the calculations,
Line 64 in the abstract not in the introduction any comparation of „howing an expansion of ultra-processed“ evidence is shown
In +the rest of the text nothing is related to this author
Line 67 the same problem the litearture about „The a fact that could be associated to increasing levels of noncommunicable diseases“ is not related to your reasersh and results.
Line 69 you have no description or results about the diet in your research and cannot claim comparation or findings of „patterns, which are based on fresh and perishable whole or minimally processed foods, some of which are orphan crops“
Line 71 you have no evidence in your researchor comparation to the following „ The consumption of these products is more suitable socially, environmentally and nutritionally.“
And so on
Line 246 „ no data are shown in the paper about „The survey collected information on household characteristics, housing conditions, education, general health characteristics, nutrition, household income and credit, household transfers, information communication technology, domestic tourism, shocks to household welfare and access to justice“ and they used 47 countries ????
These evidence is used in the calculations !!!
No evidence is given about the number of rural and urban population ??? (literature from other authors)
What is the definition of both groups less/more aflluent– statistically and in your text(literature from other authors).
What is the definition of less /more afluent
The picture in line 275 should be revised, you canot put
Orphan crop appart because it consist of a group of agricultural crops
Rice
Maize
Wheat
Cereals and pulses Fortified flour – this is also a group of food products and cannot be set this way
Breakfast cereals– this is also a group of food products and cannot be set this way
Pulses and nuts– this is also a group of food products and cannot be set this way
Bread and cakes– this is also a group of food products and cannot be set this way
Pasta– this is also a group of food products and cannot be set this way
TABLE 1.
Out of the sky the term urban poor comes out from the calculation,
It was not stated in the introduction, not oin the literature not from other author findings ??
CONCLUSION
Should be tottaly revised
Starting from the first sentence in line 358 „The purpose of this paper has been to assess the potential impact, in terms of food choices and nutrition,…“
totally wrong
this is not shown in the text, not stated as purpose
the purpose that you have shown was the difference of orphan crops use among the rural and urban population and differences noticed
line 363
in your findings you have shown no evidence about environmental impacts
Reviewer 2 Report
The processed paper has been analyzing an interesting topic. The idea of the paper is attractive. The research devoted to orphan crops is not so popular as it is in the case of staple food items, but for many poor developing countries orphan crops are very important for the stability of their food market and also for local food security. The paper is suffering because of several following problems:
First, the idea of the paper is properly introduced. The importance of the topic is not well highlighted.
Second, the processed literature overview is very poor and very limited. The real level of knowledge in area under the investigation is not described. The existing knowledge gap is not specified and highlighted.
Third, the character of objective is too general one.
Fourth, however the methodology part of the par is quite extensive and processed in detail - the results part of the paper is very poor. Individual results and findings are not explained. There is no relevant discussion related to individual outcomes. Individual findings are not discussed in relation to any other authors' findings. Differences existing among individual groups of households are not discussed and explained.
Fifth, final conclusion could considered as disappointment. It is too poor and no really valuable outputs are presented. The originality of results presentation is limited. The limits of the research are not mentioned. Also there is no point related to the future research activities. For whom the processed paper could be attractive?
The paper should be significantly revised.
Reviewer 3 Report
The paper is well done
Round 2
Reviewer 1 Report
Dear Authors,
I have made clear about this publication that it was already published in a conference Proceedings with the exact text that is offered to Sustainability journal.
I have recieved no answer about this issue but this is plagiarism, a scientist cannot publish twice the same text.
In order to publish a scientific article scientistic should
1. be original
and be franc about their work and in the case of tis paper the first law of science was not respected.
2. the research shold be replicable
the paper dose not state clearly th emethodology, again I relate to orphan crops useds as a group and then compared to ecah and every product
3. newer published before
this paper not only uses teh same words but it is tottaly equal to the one published in proceedings,
therefore each and every paragraph and sentence must be changed,
the tables, formulas and figures should clearly state that they came from proceedings
Then when these changes are done we can discuss each revision made.
Reviewer 2 Report
The rewritten paper is clearer. The first three problems are solved and already acceptable. Unfortunately, the following two problems are still existing:
However, the methodology part of the paper is quite extensive and processed in detail - the results part of the paper is still limited. Individual results and findings are not explained in deep. Still, there is no relevant discussion related to individual outcomes and their importance. Individual findings are not discussed in relation to any other authors' findings. Differences existing among individual groups of households are not discussed enough.
Final conclusion, however it is revised, could be still considered as rather weak. It is too poor and no really valuable outputs are presented. The originality of results presentation is limited. Why do you consider your outputs as really important and attractive to be published?
Minor revisions are still required.
Round 3
Reviewer 1 Report
Dear authors,
the comments are highlighted in the paper
